# On Limitations of the Transformer Architecture

**Binghui Peng**
Department of Computer Science
Columbia University
bp2601@columbia.edu

**Srini Narayanan**
Google DeepMind
srinin@google.com

**Christos Papadimitriou**
Department of Computer Science
Columbia University
christos@columbia.edu

## Abstract

What are the root causes of hallucinations in large language models (LLMs)? We use Communication Complexity to prove that the Transformer layer is incapable of composing functions (e.g., identify a grandparent of a person in a genealogy) if the domains of the functions are large enough; we show through examples that this inability is already empirically present when the domains are quite small. We also point out that several mathematical tasks that are at the core of the so-called compositional tasks thought to be hard for LLMs are unlikely to be solvable by Transformers, for large enough instances and assuming that certain well accepted conjectures in the field of Computational Complexity are true.

## 1 Introduction

The Transformer architecture (Vaswani et al., 2017), an otherwise singularly promising approach to AI, is known to be plagued by the problem of *hallucinations:* an answer to a user prompt is too often incompatible with the device's training data and prompt. There is now a vast literature on hallucinations and their nature, typology, and remedies, see for example the survey (Ji et al., 2023).

Are there root causes of the hallucination phenomenon that can be imputed to the Transformer architecture? Theoretical limitations of Transformers have been pointed out in the past, starting with Hahn's 2020 paper (Hahn, 2020), where it was proved that Transformers cannot recognize simple patterns such as parity (e.g., whether a phrase contains an even number of negations) or balanced parentheses. However, the elegant proofs in Hahn (2020), inspired by Complexity Theory, are asymptotic in nature, and it appears that the proven limitations take hold only for unrealistically large inputs; in fact, it has been demonstrated that there are Transformers on which these functions can be computed reliably for all practical purposes (Ebrahimi et al., 2020; Yao et al., 2021). Transformers have also been studied through Complexity Theory — an important lens for understanding the limitations of computational systems — culminating to Merrill & Sabharwal (2023b), where it was shown that, computationally, Transformers belong to a rather weak complexity class, namely logspace-uniform $\mathbf{TC}^0$; we elaborate on this point of view in Section 4. Also recently, Sanford, Hsu, and Telgarsky (Sanford et al., 2023) identified a particular mathematical problem called *3-Matching*, which cannot be computed by single-layer multi-head Transformers: Given a sequence of integers, identify three numbers in the sequence that add up to zero modulo a given large number. Very interestingly, (a) the easier problem of 2-Matching *can* be solved

by Transformers, but *cannot* be solved by feed-forward deep nets, establishing a kind of supremacy of Transformers over the rest of ML; and (b) these negative results manifest themselves for rather small prompt sizes. Note that, the conceptual contribution of this result notwithstanding, the 3-matching function identified in Sanford et al. (2023) is not a compelling example of the kinds of problems that Transformers are meant to solve. The question arises, can we identify impossible tasks that are closer to the intended use of the architecture? This is our focus here. We point out a new fundamental limitation of the Transformer architecture, namely that it has serious difficulties computing a very simple, and practically important, semantic operation we call *function composition*, explained next.

In a recent paper on the topic of hallucinations (Guan et al., 2024), the opening example is a wrong answer to the question `what is the birthday of Frédéric Chopin's father?`, when these two facts were included in the prompt: (a) `the father of Frédéric Chopin was Nicolas Chopin` and (b) `Nicolas Chopin was born on April 15, 1771`[1]. This is an example of function composition, where the functions to be composed are *birthday-of* and *father-of*. Interestingly, the aforementioned paper (Guan et al., 2024) goes on to propose retrofitting Transformers with Knowledge Graphs — precisely the right tool for performing function composition — in order to mitigate hallucinations. Another example of function composition would be, given the facts `London is in the UK`, `Alan Turing was born in London`, among other such facts, to ask: `In which country was Turing born?` Or, presented with Matthew's genealogy (Mat): `Abraham was the father of Isaac, Isaac was the the father of Jacob, ...` and so on with 39 more sentences of this type, to ask: `Did Isaac have any grandchildren?`

Besides its critical role in combining relational information in the data, function composition is also an important ingredient of language understanding, the core competence of Transformers. In the field of pragmatics, *indexicals* are words referring to entities in the context in which the utterance occurs. For example `his` in the sentence `I have a brother and his name is John`, or `this` in the sentence `this dog has style` are indexicals. Now, when in an utterance one indexical refers to another, understanding the utterance entails function composition. Consider this:

```
Transformers occasionally hallucinate, that is, they generate
output inconsistent with the training data and prompt. However,
this deficiency is rather rare, and techniques such as
post-filtering can alleviate it.
```

Understanding what is meant by the last word `it` of this utterance, one needs to compose two indexicals: First one has to recognize that the indexical `it` points to `this deficiency`, and then that `this deficiency` refers to the particular deficiency that `Transformers occasionally hallucinate`. It appears that humans have little difficulty composing idexicals — but how about Transformers?

So it seems desirable — indeed important — that LLMs perform function composition reliably. *In this paper we prove that they cannot:* function composition is an inherent weakness of the Transformer architecture. In particular, we show that a single Transformer attention layer cannot compute the answer to a function composition query correctly with significant probability of success, as long as the size $n$ of the domain of the function satisfies $n \log n > H(d+1)p$, where $d$ is the embedding dimension, $p$ is the precision, in bits, required for the calculation, and $H$ is the number of attention heads. In fact, the proof of our impossibility theorem suggests that this weakness has its roots in the nature of the softmax computation that allows the next embedding of a token to be computed with very scant non-local information.

Our impossibility result holds for a single, multi-headed attention layer; however, we suspect that the weakness that it exposes also burdens multi-layer Transformers, and in Appendix B we see anecdotal evidence that LLMs appear unable to reliably compose functions with domains that are far smaller.

---

[1]It is a tragic coincidence that the great romantic composer Frédéric Chopin was tormented by hallucinatory episodes throughout his brief life...

The *chain of thought* maneuver (CoT) (Wei et al., 2022) is known to help with the problem of hallucinations by inducing the LLM to generate prompts which break down the task in hand into smaller steps eventually leading to the correct answer. Indeed, a simple CoT scheme can plausibly mitigate our impossibility result on composition by generating a short prompt. However, we also prove a theorem implying that a Transformer layer with CoT needs far more tokens in the generated CoT prompt to solve the composition problem for a *cascade* of many compositions (Theorem 2)

We also provide a different impossibility argument that holds for any number of layers, and concerns a different genre of hallucinations: wrong answers to *compositionality tasks*. In Dziri et al. (2023), it is demonstrated through extensive experimentation that Transformers have trouble carrying out tasks that require sequential composition of elementary tasks, such as multiplying multi-digit integers and solving logical puzzles, and in fact this failure grows quickly with the required depth of composition. We point out (Theorem 3) that, under a widely accepted complexity assumption — akin to $\mathbf{P} \neq \mathbf{NP}$ albeit in the domain of memory — multi-layer Transformers are incapable of performing several elementary computations that are crucial in carrying out compositionality tasks such as those discussed in Dziri et al. (2023).

Finally, it has been often pointed out that it is the very nature of Transformers as *probabilistic language generators* that renders them likely to veer off their grounding on the training data and prompt — see for example McCoy et al. (2023), where it is demonstrated through extensive experimentation that low-probability answers (as well as low-probability inputs, or low-probability tasks) are harder for transformers to get right. In general, LLM models maximize the probability of generating the next token given the context in an auto-regressive manner, and this must continue even when there is no clear winner. That is, the LLM generates outputs even when the probability of the predicted token is low. When this intensifies, the model will generate a low-likelihood series of tokens, resulting in an unreliable output.

There are many situations when this could happen. First, as McCoy et al. (2023) point out, LLM are particularly bad when there is a low probability of the input, task, or output, even if the underlying training sequence is deterministic. Yet another case is one where the input context is under-specified or ambiguous. When the input context does not provide sufficient information for a clear and optimal token choice, the estimated probabilities obtained from applying the logit function to the softmax output of the transformer are distributed such that the difference between the highest and subsequent probabilities *is relatively small*, there is a higher chance that in the auto-regressive model, the incorrect token will be picked. This situation is also correlated to the case where the conditional entropy is likely high. In all these cases, the generation of the incorrect token is more likely; and once the sequence has an incorrect next token, there is a significant chance that this error cascades into a hallucination. In a recent paper (Kalai & Vempala, 2023), this phenomenon is studied more formally by considering the statistics of very rare patterns in the training data.

## 2 Preliminary definitions

**Transformer.** To model mathematically computation by Transformers, we adapt slightly the formal model of (Sanford et al., 2023). A *self-attention unit* is a function $\mathcal{A} : (\mathbb{R}^D)^N \mapsto (\mathbb{R}^d)^N$, where $N$ is the sequence length, $D$ is the embedding dimension, and $d$ is the output dimension. $\mathcal{A}$ is defined in terms of three real matrices $K, Q, V \in \mathbb{R}^{d \times D}$. For simplicity, we assume that the key, query, value matrices $K, Q, V$ share the same dimension. On input $X = (x_1, \ldots, x_N) \in (\mathbb{R}^D)^N$, the attention unit $\mathcal{A}$ calculates, for each $i = 1, \ldots, N$, the output

$$y_i = \sum_{j \in [N]} r_{i,j} V x_j \in \mathbb{R}^d \tag{1}$$

where the attention probability

$$(r_{i,1}, \ldots, r_{i,N}) = \text{softmax}(x_i^\top Q^\top K x_1, \ldots, x_i^\top Q^\top K x_N)$$
$$= \left( \frac{\exp(x_i^\top Q^\top K x_1)}{\sum_{j \in [N]} \exp(x_i^\top Q^\top K x_j)}, \ldots, \frac{\exp(x_i^\top Q^\top K x_N)}{\sum_{j \in [N]} \exp(x_i^\top Q^\top K x_j)} \right).$$

We assume that the computations of the self-attention unit are carried out with a precision of $p$ bits.

An *H-headed transformer layer* $\mathcal{L}$ consists of $H$ self-attention units sharing the same input, together with a combining function $\Phi$ which maps, for each $i$, the $H$ outputs of the layer to an output token in $\mathbb{R}^d$. Finally, a *Transformer* $\mathcal{T}$ is the cascade of several transformer layers.

Notice that our definition ignores certain features of the Transformer architecture, such as input embedding and pre- and post-processing of individual tokens by feed-forward networks; however, it is easy to see that input embedding and the pre-processing can be absorbed in the input tokens, while post-processing can be incorporated into the combining function $\Phi$, and therefore, this omission does not affect the validity of our argument.

**Function composition.** We next define the *function composition* problem. Consider two functions, $g$ mapping a domain $A$ to a domain $B$, and $f$ mapping $B$ to another domain $C$ — for example, $g(a)$ could be the mother of person $a \in A$, and $f(b)$ is the profession of person $b \in B$. These functions will be described in a prompt $X$. The $N$ tokens of $X$ are divided into three parts:

- **Part 1.** The first part describes the function $g$ through $|A|$ sentences in simple, unambiguous language separated by punctuation, e.g. `the mother of John is Helen; the mother of Charlotte is Eve;` etc.;

- **Part 2.** Similarly, the second part consists of $|B|$ sentences describing the function $f$, e.g. `Helen is a doctor; Jane is a teacher;` etc. and

- **Part 3.** The query, e.g. `what is the profession of John's mother?`[2]

Notice that the number $N$ of input tokens is a small multiple of the domain size of the functions. We say that an $H$-headed Transformer layer $\mathcal{L}$ computes the function composition correctly if, for any input prompt in the correct format, the output of the layer corresponding to the token of the query (to the token ? or potentially <eos> in this example) is the correct answer of the composition query.

In the body of the paper we also introduce similar tasks, such as iterated function composition and reachability, whose definitions are simple extensions of the above.

**Information theory.** We use standard notation from information theory. If $X, Y, Z$ are random variables, $H(X)$ is the entropy of $X$ and $I(X; Y)$ is the mutual information between $X$ and $Y$. We write $\ln(\cdot)$ for the natural logarithm and $\log(\cdot)$ for base two.

## 3 The Impossibility of Composition

We prove the following:

**Theorem 1.** *Consider a function composition problem with input domain size $|A| = |B| = |C| = n$, and an H-headed transformer layer $\mathcal{L}$ with embedding dimension $d$ and computation precision $p$, and assume that $H(d+1)p < n \log n$. Then $\mathcal{L}$ cannot solve correctly the function composition problem. In particular, if $R = n \log n - H(d+1)p > 0$, then the probability, over all possible functions and queries, that $\mathcal{L}$ answers the query incorrectly is at least $\frac{R}{3n \log n}$.*

The proof relies on Communication Complexity (Kushilevitz & Nisan, 1996), a subfield of Complexity Theory in which one measures the number of bits that need to be exchanged

---

[2]It is fine to think that these sentences come in this order, but the proof does not require it.

between distributed computational agents possessing different parts of the input, in order for one of them to obtain the result of the computation. The agents are assumed to have unbounded computational capabilities, and to be restricted only in terms of their communication. One important classical result in Communication Complexity is the communication difficulty of the *set disjointness problem:* If Bob and Alice are each given a vector of $n$ bits and they want to compute the Boolean inner product of these bit vectors — that is, to tell if there is an index $i$ such that the $i$th bit of both Bob and Alice is a 1 — then they must communicate $n$ bits. In fact, this result used is in the impossibility proofs in Sanford et al. (2023). Another classical problem in this field is *pointer chasing:* Alice and Bob are given two functions $A$ and $B$, respectively, from $[n]$ to $[n]$, and they need to compute a composition of these functions, say $A(B(A(B(A(0)))))$. Compositions can obviously be computed by the agents alternatingly communicating $\log n$ bits to each other: in this example, Alice communicates to Bob the value of $A(0)$, then Bob tells Alice the value of $B(A(0))$, and so on. But what if one less rounds of communication is allowed? Or, if the same number of rounds is allowed, but Bob must start? Over the four decades since this problem was first posed (Papadimitriou & Sipser, 1982), it has been shown that, if one less round of communication is desired, or if Bob has to start the communication, then exponentially more bits must be exchanged. These lower bounds have been used to inform many fields of Complexity, including lower bounds of bounded-depth circuits, see for example Kushilevitz & Nisan (1996), and even the complexity of Machine Learning (Chen et al., 2022).

Here we use a slight variant of this classical framework. We assume three agents, whom we call *Faye, Grace, and Xavier*. Faye knows a function $f$ from $[n]$ to $[n]$, Grace knows another such function $g$, and Xavier knows a number $x \in [n]$. We can actually assume that Faye and Grace both know Xavier's value $x$. The only communication allowed is from Faye and Grace to Xavier — *not* between Faye and Grace. Our goal is for Xavier to know the value $f(g(x))$ so that Faye communicates to Xavier as few bits as possible. Notice that we do not restrict the number of bits Grace can communicate to Xavier.

**Lemma 1.** *If fewer than $n \log n$ bits are communicated by Faye to Xavier, then Xavier cannot know the value $f(g(x))$. In particular, if only $n \log n - R$ bits are communicated from Faye to Xavier for some $R > 0$, then the probability, over all pairs of functions, that the composition is computed incorrectly is at least $\frac{R}{3n \log n}$.*

The proof can be found at Appendix A

**Remark 1.** *The lower bound on the error probability in the statement of the Lemma, $\frac{R}{3n \log n}$, is optimal within a small constant factor. In proof, an upper bound of $\frac{1}{n} \lceil \frac{R}{\log n} \rceil$ is possible, by the following construction (we assume $n$ is a power of 2 for simplicity): Faye sends to Xavier $n \log n - R$ bits, by which she encodes the the first $n - \lceil \frac{R}{\log n} \rceil$ values of the function. With this scheme, an error happens if one of the last $\lceil \frac{R}{\log n} \rceil$ values of the function is queried, and answered the wrong way. This probability is at most $\frac{1}{n} \lceil \frac{R}{\log n} \rceil$, which is just a factor of three greater than the lower bound.*

We now turn to the proof of the theorem.

*Proof.* For the purposes of contradiction, suppose that there is a self-attention layer $\mathcal{L}$ that can reliably combine any two functions $f$ and $g$ on any domains $A, B, C$ of size $n$, such that $n \log n > H(d + 1)p$. We will show that this contradicts Lemma 1.

Suppose that the three agents Faye, Grace, and Xavier are as in the lemma, and they wish to compute $f(g(x))$; we claim that they could do so by using $\mathcal{L}$ as follows: They put together the prompt of a function composition problem to be solved by $\mathcal{L}$, where Faye supplies the token strings associated with function $f$ — say, `the value of f applied to 0 is 3`, etc. Grace supplies similarly the tokens associated with function $g$, and Xavier supplies the query part: `what is the result of f applied to g applied to 23?` — recall that Xavier knows $x$, and it happens to be 23. We assume the token corresponding to 23 is token $t$, that is, $x_t = 23$. Then the three agents compute the result of the computation of $\mathcal{L}$ that corresponds to the token $t$, as explained below. Recall that, by our assumption that $\mathcal{L}$ performs function composition, this result must be the required answer $f(g(x))$.

The three agents communicate to compute, for each head, the final embedding that corresponds to the token $t$, and then Xavier applies the finishing function $\Phi$ to compute the final result, which will be the answer of the composition problem. For each head, the result at token $N$ can be written as

$$y_t = \frac{\sum_{j=1}^N r_{t,j} V x_j}{\sum_{j=1}^N r_{t,j}}, \text{where } r_{t,j} = \exp(x_t^\top Q^\top K x_j). \tag{4}$$

The key observation now is that this expression can be written as $y_t = \frac{F+G+X}{F'+G'+X'}$, where $F$ is the part of the numerator that corresponds to Faye's tokens $x_j$, similarly $G$ corresponds to Grace's tokens, $X$ corresponds to Xavier's tokens, and similarly for the denominator. Hence, Faye can compute and communicate to Xavier quantities $F$ and $F'$, and similarly for Grace and $G, G'$; then Xavier can add to these the terms $X$ and $X'$, divide, and thus compute $y_N$. Repeating for all heads and combining with $\Phi$, Xavier can compute $f(g(x))$ and obtain the desired answer to the composition problem. But now notice that this was accomplished through a communication of only $H(d+1)p$ bits from Faye to Xavier — $dp$ bits for $F$ and $p$ bits for $F'$ for each of the $H$ heads. By hypothesis, this quantity is less than $n \log n$, contradicting Lemma 1. The second part on error probability follows from the same reduction, completing the proof. $\square$

**Remark 2.** *The proof of the probabilistic statement assumes that $f$ is a uniformly random function from $[n]$ to $[n]$. To prove a negative result, some probabilistic assumption is necessary; for example, if $f(x)$ happens to be a for all $x$, far fewer bits need be communicated from Faye to Xavier. The statement can be extended to cover more general distributions, but then the entropy of $f$ would replace $n \log n$ in the denominator of the statement.*

**Remark 3.** *It is clear from the proof that function g plays a minimal role in the lower bound, since its value on $x$ is communicated for free to Xavier. Indeed, one can argue that what is proved impossible for the Transformer layer is the* evaluation *of a function $f$. The reasoning in the proof of the Theorem can be repeated for prompts which, after listing the birthplaces of many luminaries end like this:* ``...where was Einstein born?'' *, as long as the answer must appear at the position of the last token.*

**Chain of Thought**

Can CoT help solve the composition problem? Intuitively, the answer is "yes." For any composition problem — for example, the prompt about Turing, London, and England in the introduction — we can help the LLM successfully answer the question ``In which country was Turing born?'' by generating a short CoT prompt that breaks the question into simpler ones, e.g. ``Let's see, Turing was born in GENERATE, and GENERATE is in the country of GENERATE, so Turing was born in GENERATE.'' However, we prove below that an arbitrarily large number of CoT steps are needed to solve the generalization of composition to *many* function applications. In the *iterated function composition problem* we are given $K$ functions $f_1, f_2, \ldots, f_K$, and we need to calculate $f_K(f_{k-1}(\ldots(f_1(x))))$ for $x \in [n]$. In fact, in our proof we shall consider $f^{(K)}(x) = f(f(\ldots f(x)))$ — that is, we shall take $f_1 = \cdots = f_K$.

**Theorem 2.** *Let $H$ be the number of attention heads, $d$ the embedding dimension, $p$ the computation precision, and $n$ be the domain size of the iterated composition problem. A Transformer layer requires $\Omega(\sqrt{\frac{n}{Hdp}})$ CoT steps for answering correctly iterated function composition prompts.*

*Proof.* We reduce from another classical problem in Communication Complexity called *pointer chasing*. Let $n$ and $c$ be two positive integers. In the $(n, c)$-pointer chasing problem, Alice knows a function $f_A : [n] \to [n]$ and Bob knows another function $f_B : [n] \to [n]$. The pointers $w^{(1)}, w^{(2)}, \ldots$ are recursively defined as

$$w^{(1)} = 1, \quad w^{(2)} = f_A(w^{(1)}), \quad w^{(3)} = f_B(w^{(2)}), \quad w^{(4)} = f_A(w^{(3)}), \quad w^{(5)} = f_B(w^{(4)}), \quad \ldots$$

The communication proceeds for $2r$ rounds, with Alice starting. The goal is for Bob to output the binary value of $w^{(2r+2)}$ (mod 2). The following summarizes what is known about this problem:

**Lemma 2** (Nisan & Widgerson (1991); Klauck (2000); Yehudayoff (2020))**.** *Any randomized protocol for the pointer chasing problem with error probability at most* $1/3$ *under the uniform distribution must involve the transmission of at least* $n/(2000c) - 2c \log n$.

The connection between pointer chasing and CoT is summarized below, the proof can be found at Appendix A

**Lemma 3.** *For any* $K \geq 1$, *suppose there is a Transformer layer* $\mathcal{L}$ *with $H$ attention heads, dimension $d$, and precision $p$ that solves the $K$-iterated function composition within $R$ CoT steps, then there is a communication protocol for* $(n, K-1)$-*pointer chasing, communicates in $2R$ rounds and exchanges* $2RH(d+1)p$ *bits.*

Combining Lemma 2 and Lemma 3, and taking $K = \frac{1}{100}\sqrt{\frac{n}{Hdp}}$, we complete the proof. $\square$

## 4 Compositionality and Logarithmic Space

In a recent paper (Dziri et al., 2023) a novel genre of hallucinations was identified: extensive experimental evidence is presented that Transformers perform poorly on *compositional* tasks, that is, tasks requiring the repeated iteration of elementary tasks; similar phenomena have been observed elsewhere (Merrill & Sabharwal, 2023b; Feng et al., 2023; Merrill & Sabharwal, 2023a). The main examples of compositional[3] tasks explored in Dziri et al. (2023) are:

- multiplication of multi-digit integers modeled as an arithmetic circuit with single-digit values and inputs;
- a simple sum maximization problem over a sequence of integers under the restriction that two successive integers cannot be both added to the sum; this again can be reduced through dynamic programming to an arithmetic circuit with plus and max gates; and
- Logic puzzles such as "Einstein's Riddle"[4]

Wrong answers to large families of simple questions such as these constitute a special category of hallucinations, and it is of interest to explore its causes. It turns out that, to do so, we must turn the page of our negative results of the previous section and Communication Complexity arguments, and employ the theory of Computational Complexity (Papadimitriou, 1993; Arora & Barak, 2009) to study certain basic computational problems underlying the tasks studied in Dziri et al. (2023):

*Circuit evaluation:* Given the description of a circuit with gates, which can be either Boolean or arithmetic operations, as well as the values of all input gates of the circuit, evaluate the output(s) of the circuit. Multiplying decimal integers with multiple digits is an example of such a circuit; solving the adjacency-restricted largest sum problem of Dziri et al. (2023) is also the evaluation of a circuit, this time with $+$ and *max* gates.

*Derivability* is yet another generalization of our composition task of the previous section which we believe captures many aspects of the informal notion of compositionality. We are given a finite domain $S$ — intuitively, the partial solutions of the problem in hand — and a relation $D \subseteq S \times S$ — intuitively, legal one-step derivations. We are also given two subsets

---

[3]We note here that, despite the obvious linguistic affinity of the two terms "composition" studied in the previous section and the "compositionality" of Dziri et al. (2023), the two terms are further than they seem. Compositionality is an informal category that is vastly more general than composition, which is a specific mathematical concept.

[4]https://en.wikipedia.org/wiki/Zebra_Puzzle

of $S$, $I$ (for initial partial solutions) and $F$ (for final partial solutions). The question is: are there elements $a_1, a_2, \ldots, a_k \in S$ such that (a) $a_0 \in I$; (b) $a_k \in F$, and (c) for all $j$ such that $0 < j \leq k$, $D(a_{j-1}, a_j)$?

***Logical reasoning:*** Logic puzzles can be typically formulated as instances of satisfiability (or SAT). This problem is NP-complete and, even though large instances arising in practice can be solved by sophisticated techniques developed over decades of research, it would not be surprising if LLMs may have problems in dealing with arbitrary logical expressions. There are, however, three tractable special cases of SAT that underlie much of tractable common-sense reasoning: *2-SAT*, the satisfiability of CNF formulas with two literals in each clause; *Horn SAT,* the corresponding problem for Horn clauses, that is, clauses with at most one positive literal; and *Mod 2 SAT,* the solution of a system of linear equations modulo 2. Note that these are the only nontrivial tractable special cases of SAT. Can we expect LLMs to handle them?

We point out below that, assuming certain well accepted conjectures in Computational Complexity, all of these tasks are impossible for a multi-layer Transformer to perform reliably and for large enough prompts. We start with the following:

**Theorem 3.** *The computation of of an L-layer Transformer on a prompt of length N can be carried out with $O(L \log(n))$ bits of memory, when the precision, the number of heads and the embedding dimension is polynomially bounded $p, H, D \leq \text{poly}(N)$.*

*Proof.* The Observation is not hard to verify for a single Transformer layer: In each head of each layer of a Transformer, one only has to compute the next layer embedding $y_i$ for each token, through the formula in Equation (1). This formula can be computed by a program with two loops over $i$ and $j$, both ranging over all $N$ tokens. These loops nest three more loops over $d, D$ and the Taylor iteration for computing the exponentiation function. Since the precision of all numbers involved is logarithmic with respect to the problem size $N$ (the prompt length), everything else, including arithmetic operations, can be carried out with $\log(N)$ memory.

This observation can be made more formal by defining a Turing machine with an input tape containing the data of the computation (the $N$ tokens and the entries of the matrices $Q, K, V$), as well as six work tapes: Two for maintaining the indices $i$ and $j$, three for the indices ranging over $[d]$ and $[D]$ and one for the partial results of the arithmetic operations; the work tapes, taken together, use a total number of bits that is $O(\log N)$. When there are multiple attention heads, as long as the combination function $\Phi$ can be carried out also in log-space (which is true for both finite-depth neural networks as well as matrix-vector multiplication), the output at each token can be computed in logarithmic space.

So far we have argued that, given access to the input tokens and the attention matrices, the computation of the first layer can be carried out in logarithmic space. To extend this to two layers, recall that the computation of the second layer is just like that of the first, except that access to the outputs $y_i$ of the first layer is needed, instead of the input tokens. This can also be performed in $O(\log N)$ space by *recomputing*, in a separate set of tapes, the $y_i$ values one by one as needed by the computation of the second layer — incidentally, this recomputation maneuver is a common way of saving memory by exploiting a memory-time trade-off. In the same way, any constant number of layers can be computed in logarithmic space, through nested loops. The number of layers will appear as a multiplicative constant of the space requirement of the computation, as well as in the *exponent* of the time requirement. □

It follows that the computation of a constant layer Transformer belongs in the complexity class **L** (standing for *logarithmic space* (Papadimitriou, 1993; Arora & Barak, 2009)).

**Observation 1.** *The computation of a constant layer Transformer on a prompt of length N can be carried out with $O(\log N)$ bits of memory.*

This is not a new result; it has been recently established in Merrill & Sabharwal (2023b) that the computation of Transformers lies in the complexity class log-uniform $\textbf{TC}^0$, which is more

restrictive than logarithmic memory; in fact, the implications of this result for the capabilities of Transformers are also briefly discussed in Merrill & Sabharwal (2023b). However, we feel that the simple arguments we present here, both for proving the observation and point out its connections to compositionality tasks, are simple, intuitive and useful.

Now, next to the paramount $\mathbf{P} \neq \mathbf{NP}$ conjecture about time complexity, there is another important, classical, and also broadly accepted analogous conjecture about memory: $\mathbf{L} \neq \mathbf{NL}$. It states that nondeterministic logarithmic memory is more powerful than its deterministic counterpart (Papadimitriou, 1993; Arora & Barak, 2009). Just as $\mathbf{NP}$ has complete problems, such as SAT, which witness its conjectured difference from $\mathbf{P}$, there are $\mathbf{NL}$-complete problems, and two of the best known among them are 2-SAT and Derivability (classically known as *Reachability*). Circuit evaluation and Horn-SAT are even harder: they are both complete for $\mathbf{P}$, which includes $\mathbf{NL}$.

We summarize the previous discussion as follows:

**Observation 2.** *The four problems of Derivability, 2-SAT, Horn SAT, and Circuit evaluation cannot be solved by multi-layer Transformers unless $\mathbf{L} = \mathbf{NL}$. In fact, for the latter two problems the result is true unless the stronger statement $\mathbf{L} = \mathbf{P}$ holds. For Mod 2 SAT, the result is true unless the weaker statement $\mathbf{L} = \mathbf{Mod\ 2\ L}$ holds.*

We believe that this theorem goes a long way towards explaining the shortcomings of Transformers identified in Dziri et al. (2023), given the affinity between the problems proved impossible above and the tasks studied in that paper. Importantly, it is demonstrated experimentally in Dziri et al. (2023) that the performance of Transformers on these tasks deteriorates rapidly as the *depth* of the task increases. This is in good agreement with our complexity explanation, because the relevant complexity results kick in only when the depth is larger than $\log N$ — for compositional tasks of depth smaller than that, it is not hard to see that logarithmic memory is enough.

## 5  Discussion

We used complexity arguments of two different genres — Communication Complexity and Computational Complexity — in order to elucidate certain shortcomings of the transformer architecture, namely an *information bottleneck* as well as difficulties in dealing with *depth*. We showed that the elementary function composition problem cannot be solved by a single Transformer layer, and that CoT can solve the iterated composition problem only by generating a prompt that has length $\Omega(\sqrt{N})$. These mathematical results are limited in two ways: (a) they take hold when the domain size is larger than the dimension parameters (which are typically in the hundreds), and (b) they break down for multiple layers. We also provide evidence from Complexity Theory — of the classical Turing machine variety — that the compositionality tasks known empirically to be hard for Transformers contain computational ingredients and primitives that are impossible for Transformers to deal with.

The reader is reminded that the complexity arguments we employ here come with caveats. The impossibility result for composition holds for a single layer, in a probabilistic way (the error probability is nonzero but not one), and only when the domain of the functions is larger than the parameters of the Transformer layer. The results based on Computational Complexity come with a different set of caveats: They hold only if certain yet unproven, if widely accepted, conjectures are true, and even then they are asymptotic, holding for instances larger than an unknown size, for which the assumed asymptotic conjectures take effect. Complexity results such as these are mere warnings that these problems have a *fundamental incompatibility* with the Transformer architecture, and therefore one should not expect that these problems to be solvable in practice *ad infinitum.* However, as with other complexity conjectures such as $\mathbf{P} \neq \mathbf{NP}$, it is often the case that computational difficulties start happening for instances of reasonably small size. For example, we already know (Dziri et al., 2023) that Transformers have difficulties with compositionality tasks of rather small sizes, and in Appendix B we present anecdotal evidence that LLMs often respond erroneously to prompts of small sizes related to function composition. Naturally, the opposite phenomenon is also common: some $\mathbf{NP}$-complete problems such as 3SAT

seem to be amenable to efficient solutions in practice for very large instances; however, this is typically accomplished by extensive exploration over decades of a large arsenal of algorithmic techniques relying on the kinds of instances that appear in practice, and not through a fixed algorithmic architecture.

The real tragedy of complexity lower bounds is not their tentative and asymptotic nature; it is that (a) they are rare and hard to come by, and they come in very few known kinds; and (b) they tend to be overly conservative, in that they often vastly overestimate the capabilities of the computational agents they are trying to delimit. Lemma 1 is a good example: it is designed to hold even if Grace uses the most sophisticated math to encode her input into her message. But when the lemma is applied in Theorem 1, Grace is very restricted, in that her message to Xavier is not a clever coding of her tokens, but the two particular simple numerical expressions that we call $G$ and $G'$ in the proof. It is intuitively clear that this computation — the token values projected, exponentiated, and then cross-multiplied and averaged — is a very poor way to encode the $n$ values of function $g$ so that Xavier can recover each one of them readily. This observation highlights an interesting open problem, the opportunity to develop a more sophisticated variant of Communication Complexity for computationally restricted agents in order to study the limitations of devices such as the Transformer.

Finally, our two genres of negative results suggest an intriguing challenge: What would it take to design a different attention layer that is immune to these two lower bound techniques, while maintaining the architecture's efficiency and effectiveness in practice? Our proof suggests a version of the softmax computation in the attention layer that is either not commutative or not associative, or one that requires more than logarithmic space. But, of course, simply evading a lower bound technique does not guarantee a tangible improvement in performance...

## Acknowledgement

We are grateful to Fernando Pereira for his insights, engagement, and constructive critiques throughout this project. Thanks also to Olivier Bousquet for many insightful comments on an earlier draft. This work was done when the third author was a visiting scientist at Google DeepMind in Zurich. The work of the first and third authors was partially supported by an NSF grant.

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

## A   Missing proof from Section 3

*Proof of Lemma 1.* The proof of the first statement of the lemma is elementary. Since Faye and Grace cannot communicate, and communication from Grace to Xavier is free, we may assume that Grace communicates to Xavier the value of $g(x)$ — intuitively, the only information he needs from her. Even though progress seems to have been made, Xavier must now apply $f$ to $g(x)$, and he knows nothing about $f$. This difficulty must be solved through communication from Faye to Xavier, as further communication from Grace obviously cannot help. There are $n^n$ possible functions $f$, and to describe the actual one completely Faye needs to send Xavier at least $n \log n$ bits, the logarithm base two of $n^n$ — in fact $n \lceil \log n \rceil$ bits suffice, because Faye can send to Xavier the values $f(0), f(1), ...$ in this order. If Faye sends to Xavier fewer than $n \log n$ bits, say $B$ bits, the precise bitstring she sends is determined by the function $f$ she knows, and thus this communication subdivides the $n^n$ candidates for the $f$ function into $2^B$ categories — one for each value of the $B$ bits sent by Faye to Xavier. Since $B < n \log n$, we have that $2^B < n^n$, and hence, by the pigeonhole principle, at least two different candidate functions $f$ and $f'$ result in the same communication from Faye to Xavier, and Xavier has no way of knowing which of the two functions is the correct one. These two functions must differ in at least one value, say $f(z) \neq f'(z)$. But what if it so happens that $g(x) = z$? In this case, Xavier has no way of knowing the correct answer to the problem he is supposed to solve, completing the proof of the first statement.

The proof of the second, quantitative, statement is more technical and relies on Information Theory and mutual information[5]. Recall the input of Grace is a random mapping $g : [n] \mapsto [n]$, for any fixed input of Xavier $x \in [n]$. Let $i^* = g(x) \in [n]$, then we know that $i^*$ is uniform over $[n]$. Let $\Pi$ be the message sent from Faye to Xavier, it has $n \log(n) - R$ bits. We first bound the mutual information between $\Pi$ and $f(i^*) = f(g(x))$:

$$
\begin{aligned}
I(\Pi; f(i^*)|i^*) &= \sum_{i=1}^{n} \Pr[i^* = i] \cdot I(\Pi; f(i^*)|i^* = i) \\
&= \frac{1}{n} \sum_{i=1}^{n} I(\Pi; f(i)) \\
&\leq \frac{1}{n} \sum_{i=1}^{n} I(\Pi; f(i)|(f(j))_{j<i}) \\
&= \frac{1}{n} \sum_{i=1}^{n} I(\Pi; f(1), \ldots, f(n)) \\
&\leq \frac{|\Pi|}{n} = \log n - \frac{R}{n}.
\end{aligned}
\tag{2}
$$

The first step follows from the definition of conditional mutual information. The second step follows from the facts that $i^*$ is uniform over $[n]$ and $\Pi$ is independent of $i^*$. The third step follows from $(f(j))_{j<i}$ are independent of $f(i)$, and the fourth step is the chain rule.

Notice that Xavier's output is just a post-processing of $\Pi$ and $i^*$, and denote by $\delta$ its error probability. Then by Fano's Inequality, we have

$$
\begin{aligned}
H(\delta) + \delta \log(n) &\geq H(f(i^*)|\Pi, i^*) = H(f(i^*)|i^*) - I(\Pi; f(i^*)|i^*) \\
&\geq \log n - (\log n - \frac{R}{n}) = \frac{R}{n}.
\end{aligned}
\tag{3}
$$

Here the first step follows from Fano's inequality, the third step follows from Eq. (2).

From Eq. (3), we can derive that $\delta \geq \frac{R}{3n \log n}$. □

We next prove Lemma 3

---

[5]Strictly speaking, the second statement implies the first; however, the simple proof of the first statement is included for reasons of exposition.

*Proof of Lemma 3.* Recall that in the pointer chasing problem, Alice receives a function $f_A : [n] \mapsto [n]$ and Bob receives a function $f_B : [n] \mapsto [n]$. Define a single mapping $f : [2n] \to [2n]$ such that

$$f(i) = \begin{cases} f_A(i) + n & i \in [n] \\ f_B(i - n) & i \in [n + 1 : 2n] \end{cases}$$

For any $i \in [n]$, we have that $f^{(k)}(i) = (f_B \circ f_A)^{(k)}(i)$ holds for any integer $k \geq 1$.

Suppose there is a Transformer layer $\mathcal{L}$ that solves the $K$-iterated function composition problem using $R$ CoT steps; we shall construct a communication protocol for pointer chasing based on it. Alice and Bob put together the function composition task for $\mathcal{L}$, where Alice supplies the description of $f(1), \ldots, f(n)$, using her knowledge of $f_A$, and Bob supplies the description of $f(n + 1), \ldots, f(2n)$, using his knowledge of $f_B$. For simplicity, we assume the $i$-th input token, $x_i$, contains the information of $f(i)$, for any $i \in [2n]$; the query $x_{2n+1}$ appears at position $2n + 1$ and it asks for $f^{(K)}(1)$. Let $K_h, Q_h, V_h$ be the key, query, value matrices of the $h$-th attention head.

The communication protocol proceeds in $2R$ rounds, where the $2r - 1, 2r$ rounds simulate the $r$-th step of CoT. Formally, for $r = 1, 2, \ldots, R$, the protocol proceeds as follows:

- At round $2r - 1$, for each attention head $h \in [H]$, Alice computes $\sum_{i \in [n]} \exp(x_{2n+r}^\top Q_h^\top K_h x_i) \in \mathbb{R}$ and $\sum_{i \in [n]} (x_{2n+r}^\top Q_h^\top K_h x_i) V_h x_i \in \mathbb{R}^d$, and sends them to Bob;

- At round $2r$, for each attention head $h \in [H]$, Bob computes $\sum_{i \in [n+1:2n]} \exp(x_{2n+r}^\top Q_h^\top K_h x_i) \in \mathbb{R}$, $\sum_{i \in [n+1:2n]} \exp(x_{2n+r}^\top Q_h^\top K_h x_i) V_h x_i \in \mathbb{R}^d$ and sends them to Alice;

- Alice and Bob compute, locally, the output $y_{2n+r,h}$ ($h \in [H]$) as

$$y_{2n+r,h} = \frac{\sum_{i \in [2n]} \exp(x_{2n+r}^\top Q_h^\top K_h x_i) V_h x_i + \sum_{i \in [r]} \exp(x_{2n+r}^\top Q_h^\top K_h x_{2n+i}) V_h x_{2n+i}}{\sum_{i \in [2n]} \exp(x_{2n+r}^\top Q_h^\top K_h x_i) + \sum_{i \in [r]} \exp(x_{2n+r}^\top Q_h^\top K_h x_{2n+i})}$$

and the next token $x_{2n+r+1} = \Phi(y_{2n+r,1}, \ldots, y_{2n+r,H})$

After $2R$ rounds, Bob knows the output of $R$-fold CoT, and can compute $f^{(K)}(1) = (f_B \circ f_A)^{(K)}(1)$, this resolves the $(n, K - 1)$-pointer chasing task. The total number of bits communicated are $2R \cdot H(d + 1)p$, as required by the lemma. □

# B  Examples

We show here a few qualitative results that illustrate the difficulty of composition for state-of-the-art LLMs. A full empirical exposition for the case of compositional failures can be found in Dziri et al. (2023). The experiments here are conducted on ChatGPT 3.5, ChatGPT 4 and Bard. The prompts used involve simple composition over spatial, temporal or relationship relations. The experiments were performed before March 26th, 2024. We allow all three LLMs to perform step-by-step reasoning, and we only show their final answer. The answers displayed are typical.

**Spatial composition**  When the prompt involves spatial information, transformer based systems appear to have problems with composition, see examples in Figure 1.

**Temporal composition**  Figure 2 shows two simple cases where temporal composition fails and all the state-of-the-art models return incorrect answers.

**Relationship composition**  We also see incorrect and hallucinatory answers when the prompt involves the composition of relationship information, see the two examples in Figure 3.

**Prompt:** Fayes is to the west of Xaive, Jill is to the north of Ken, Fayes is to the south of Ken, where is Ken with respect to Xaive?

**GPT 3.5:** East

**GPT 4:** Northeast

**Bard:** Not enough information

**Correct answer**: Northwest

---

**Prompt:** If Amy is to the southwest of Ben, Cindy is to the northeast of Amy and directly north of Ben, is Amy further from Ben or Cindy?

**GPT 3.5:** Ben

**GPT 4:** Ben

**Bard:** Ben

**Correct answer:** Cindy

Figure 1: Spatial composition produces incorrect answers

---

**Prompt:** Jan's birthday is one year after Nancy, Nancy is older than John by seven years. What's the age different between Jan and John?

**GPT 3.5:** 8 days

**GPT 4:** 8 years

**Bard:** 8 years

**Correct answer:** 6 years

---

**Prompt:** Alice is the younger sister of Bob, Bob is the elder brother of Tim. Is Alice younger than Tim?

**GPT 3.5:** Yes

**GPT 4:** Not enough information

**Bard:** Yes

**Correct answer:** Not enough information

Figure 2: Hallucinations in temporal composition

**Prompt:** Aig is the son of Bef, Caf is the son of Aig. Does Aig have any grandchildren?

**GPT 3.5:** Yes

**GPT 4:** Yes

**Bard:** Yes

**Correct answer:** Not enough information.

---

**Prompt:** Aya is the father of Bob, Charlie is the father of Cindy, Bob is the mother of Cindy. Does Aya have a grandchild?

**GPT 3.5:** Not enough information

**GPT 4:** Yes

**Bard:** No

**Correct answer:** Yes

Figure 3: Hallucinations in relationship composition

