# OpenReview forum: "On Limitations of the Transformer Architecture"
_colmweb.org/COLM/2024/Conference — COLM_

### Official Review · Reviewer_qKRW · 2024-05-10

**Rating:** 4
**Confidence:** 4
**Ethics Flag:** 1

**Summary:**

This paper presents three theoretical results:

Theorem 1. Given a string encoding of two functions f and g, and a point x, a single transformer layer cannot compute f(g(x)).

Theorem 2. Given a string encoding of a function, a number K, and a point x, a single transformer layer requires more than O(sqrt(n/Hdp)) intermediate steps (chain-of-thought) to compute f^K(x).

Observation 1. Transformer encoders are in L (log-space).

**Questions To Authors:**

In the definition of the pointer-chasing problem, are c and r the same thing?

I'd suggest changing the title to reveal more of the contents of the paper; for example, mentioning function composition or communication complexity.

The introduction's last two paragraphs don't appear to have anything to do with the rest of the paper, and end rather abruptly. Was this an editing problem?

**Reasons To Accept:**

These results are interesting, have a clear connection with problems that we'd like language models to solve in practice, and line up well with empirical observations.

**Reasons To Reject:**

Theorem 1 only deals with a single transformer layer. Although the introduction claims "it seems desirable...that LLMs perform function composition reliably. In this paper we prove that they cannot," it later says that the result only applies to a single layer and "we suspect" that the result applies to multi-layer transformers. This claim needs much more justification; after all, stacking multiple layers is just function composition, and two-layer FFNNs are much more powerful than one-layer FFNNs, so there are good reasons to suspect that adding more layers could help.

Theorem 2 gives a lower bound on the number of chain-of-thought steps needed for iterated function composition. I was a bit confused because this lower bound doesn't seem very high to me. I certainly wouldn't say that this theorem shows that an "arbitrarily large number of CoT steps are needed"!

Observation 1 is rightly labeled an observation rather than a theorem. It's already known that transformer encoders with log-precision numbers are in L-uniform TC0, indeed DLOGTIME-uniform TC0, and therefore are in L. All this is acknowledged at the top of page 8. In my opinion this whole section (4) does not need to be in the paper.

However, I'm interested in the claim that all the numbers involved require only log N precision. In Merrill and Sabharwal's results, this is part of the premise. If you have a proof that a transformer that uses arbitrary precision is equivalent to one that uses only log N precision, then this would add tremendous value to the paper.

Minor complaints:

I don't entirely agree with the claim that "asymptotic...limitations take hold only for unrealistically large inputs...these functions [parity and Dyck] can be computed reliably for all practical purposes". The cited papers are about Dyck languages, not parity, and the Yao paper is about Dyck languages with bounded depth.

Missed citation: although Merrill and Sabharwal (2023b) prove that Transformers (with log N bits of precision) are in log-space uniform TC0, a later paper ("A logic for log-precision transformers") tightened this to log-time uniform TC0.

---

> ### Author Rebuttal · Authors · 2024-05-30
>
> Many thanks for your very helpful suggestions; we address your concerns below.
>
> > Theorem 1 ..single transformer layer... adding more layers could help.
>
> Thanks for catching this. We will make this statement accurate, and we will add an explicit conjecture that no constant layer Transformer can solve the iterative function composition task (defined in the chain of thought section).
>
> > wouldn't say that this theorem shows that an "arbitrarily large number of CoT steps are needed"!
>
> Thanks for your comment. By “arbitrary large” we mean “non-constant,” in the sense that for any large constant we can find a task that requires more COT steps – we apologize for the confusion and will change it to $\sqrt{N}$ steps to clarify. We want to note that this is the first lower bound proved for Transformers with chain of thought.
>
> > Observation 1 .. It's already known that transformers ... are in L  ... proof that a transformer that uses arbitrary precision is equivalent to one that uses only log N precision.
>
> Thanks for your suggestions! Indeed Observation 1 follows from existing results, and we are very clear about that. However, we believe that our proof is simple and illuminating, and has implications for new results. For example, it follows immediately from it that Transformers with log n layers are in log^2 n space.
>
> And thanks for your suggestions on extension to arbitrary precision. If p is the precision, then our theorem could easily be extended to show that the L-layer Transformer is in space L (\log(n) + p). Hence, it gives a non-trivial characterization when p=polylog(n).
>
> For larger or even infinite p, even writing down the output takes p bits, so we need to make further assumptions in order to put it into property space complexity class (For example, if we only want to know the most significant bits of the output, then it is possible to extend our results to L \log(np), but we expect it needs more work).
>
> Finally, we note that both in theory and in practice, the bit precision is usually not the dominant term, and it is very common to assume it scales logarithmically with the problem size.
>
> Overall, we believe Observation 1 gives a simple proof for existing results and gives new results by characterizing Transformer through space complexity. We believe using space complexity to characterize Transformers is a promising direction.
>
> > Missed citation
>
> Thanks, we will add the citation.
>
> > are c and r the same thing?
>
> Yes, sorry for the typo.

---

> > ### Author Response · Authors · 2024-06-04
> >
> > We thank the reviewer for the valuable comments. As the discussion period is close to its end, we believe that we have addressed all of your concerns. We would love to hear from all of you. Please let us know if there are any remaining concerns.

---

> > > ### Author Response · Authors · 2024-06-05
> > >
> > > We thank the reviewer for the valuable comments.  As the discussion period is close to its end, we would love to hear from you. Please let us know if there are any remaining concerns.

---

> > > > ### Comment · Reviewer_qKRW · 2024-06-05
> > > >
> > > > I don’t have anything to add except for my comments above, which have not been answered. I interpret your silence to mean that in Observation 1, the assumption of log precision is indeed part of the premise and the statement needs to be corrected.

---

> > > > > ### Author Response · Authors · 2024-06-06
> > > > >
> > > > > Thank you for your response.
> > > > >
> > > > > > I interpret your silence to mean that in Observation 1, the assumption of log precision is indeed part of the premise and the statement needs to be corrected.
> > > > >
> > > > > We will change the statement of Observation 1 as follows (see our previous response for explanation).
> > > > >
> > > > > (Observation 1, new) If p is the precision, then L-layer Transformer is in space L (\log(n) + p) (assuming H, D \leq poly(n))
> > > > >
> > > > > > I don’t have anything to add except for my comments above, which have not been answered.
> > > > >
> > > > > Are there any other comments (except the log precision one) that we have not answered? Please let us know.

---

### Official Review · Reviewer_FYMs · 2024-05-11

**Rating:** 6
**Confidence:** 3
**Ethics Flag:** 1

**Summary:**

This paper applies theories from Communication and Computational Complexity to analyze the limitations of transformers architecture.

The authors specifically focus on function composition transformer architecture is inherently incapable of composing functions. The authors identify with proofs an information bottleneck: a transformer's layer capability to compute function composition becomes increasingly unreliable as the domain size increases. They also show out that the utility of CoT (which has been used historically to overcome complexity issues) is limited because the number of CoT needed also increases with domain size.

The authors also posit that multilayer transformers are incapable of reliably performing on depth-increasing tasks (e.g. logical puzzles and mutli digit multiplication) as their computation mechanism is limited to the logarithmic space.

**Questions To Authors:**

I asked the questions above.

**Reasons To Accept:**

This is largely a theoretical paper seeking to provide mathematical explanation behind the various empirical evaluations and observations of prior works (with especial focus given to the results found in Dziri et al 2023). The investigation is carefully done with in-depth discussions. I think the strength of this work is that it provides a insights into what we can expect (or more precisely, cannot expect) a LLM to be able to do, and just how much we can expect techniques like CoT can buy us in terms of improvement of the model performance in composition-heavy tasks.

I appreciated the carefully thought out limitations of the present work in the discussion, and also the readability and approachability of the writing.

**Reasons To Reject:**

Some open questions.

The impossibility result, as specified, only applies to a single multiheaded attn layer, but the authors show a few examples in the appendix that show that multi-layer might not solve the issue. To what extent has this been investigated?

The instantiations of function composition discussed in this paper (and appendix) mostly centers around the indexical or entity resolution, where functions are variants of stative relationships (is, is at ... etc). Are there other forms that function composition can take?

If inability to deal with compositionality is limited because a transformer layer simply can't hack it, then do you have any thoughts on how such limitations be overcome? Or would that mean looking beyond the current transformer architectures i we are to build models capable of dealing with such complexity.

I should also note that I defer to other reviewers on the evaluation of the specifics of the proofs and application of the theorems, as this is not within my specialty.

---

> ### Author Rebuttal · Authors · 2024-05-30
>
> We thank the reviewer for the helpful comments/suggestions; we address your questions below.
>
> >The impossibility result, as specified, only applies to a single multiheaded attn layer, but the authors show a few examples in the appendix that show that multi-layer might not solve the issue. To what extent has this been investigated?
>
> From the theoretical side,we conjecture that any constant layer Transformer could not resolve the iterated composition task (a generalization of the composition task, see the chain of thought Section for a formal definition). From the empirical side we show a few examples in which Transformers fail in very simple tasks of that sort, and we refer to (Dziri et al. 2023) for a very systematic and extensive investigation of the issue.
>
> > The instantiations of function composition discussed in this paper (and appendix) mostly centers around the indexical or entity resolution, where functions are variants of stative relationships (is, is at ... etc). Are there other forms that function composition can take?
>
> Function composition is relevant for many natural tasks, besides database lookup. For example, it is important in logical reasoning of the form ”A implies B; if A is true, then D is false; (...) We know A is true, is C true?” as well as in spatial reasoning (“Jim is to the north of Jane and to the east of Tim (…) Is Tim to the south of Elli?” and categorical reasoning (“mammals are vertebrates; apes are mammals (...)  do all apes have a cortex?”).
>
> > If inability to deal with compositionality is limited because a transformer layer simply can't hack it, then do you have any thoughts on how such limitations be overcome? Or would that mean looking beyond the current transformer architectures i we are to build models capable of dealing with such complexity.
>
> Many thanks for your question.  Indeed, we have been thinking about ways to enhance the transformer layer in exactly these directions.  One way to get around the impossibility proof of the function composition task is to change the transformer layer by computing softmax, for each token, only on the, say, largest 50% results.  To get around the impossibility of compositionality tasks, one could change the layer by having, instead of the product of the matrices Q and K, a deep net with the same number of parameters.  We can mention these ideas in our paper.

---

> > ### Author Response · Authors · 2024-06-04
> >
> > We thank the reviewer for the valuable comments. Since it is close to the end of author-reviewer discussion, please let us know if we have addressed your questions.

---

> > > ### Comment · Reviewer_FYMs · 2024-06-04
> > >
> > > Thank you for the rebuttal/response. I have considered the score, and I have decided to keep the current as is.

---

### Official Review · Reviewer_7NpS · 2024-05-11

**Rating:** 6
**Confidence:** 2
**Ethics Flag:** 1

**Summary:**

This paper uses complexity theory to argue that a) one transformer layer cannot compose functions when the domains of the functions are too large and b) that many tasks requiring compositionality (in the broad sense) are unlikely to be solvable by Transformers, assuming that $L\neq NL$.

The paper is well written, and deals with a very interesting topic and approach: examining the computational properties of Transformers. This is an essential area of research as it allows us to understand what the limitations of these ubiquitous models could be.

**Questions To Authors:**

Theorem 1 states that for  functions $g: A\rightarrow B$, $f:B\rightarrow C$ with $|A|=|B|=|C|=n$, and a Transformer layer $\mathcal{L}$ with $H$ heads, embedding dimension $d$, and precision $p$, such that $H(d+1)p < n \log n$, then $\mathcal{L}$ cannot solve the function composition problem.

A key question I had for the proof of this theorem was as follows. Part of the proof is observing that the passage of information is via the quantity $\frac{(F + G + X)}{(F' + G' + X')}$. This quantity by itself could form part of the encoding of $f(g(x))$ or of $g(f(x))$ by commutativity of addition. So for this to form part of the encoding of the desired $f(g(x))$ we need to add in positional embeddings. But presumably this will require passing on $p$ more bits of information (I must say I am not totally clear if this is true, and complexity/information theory is not really my field, so I could well be mistaken). If this is the case, can the contradiction with Lemma 1 still be derived? This may also be a problem for the proof of Lemma 3, since the order of function composition is critical there too, I think.

More general observations and questions are as follows.
1) A key concept in the paper is that of domain size. In the case of functions over the set $\{1,...,n\}$ this is well-defined. It would be good to include a discussion of what domain size means in other contexts. If the domain is the continuous interval $[1, n]$, does this mean function composition is basically impossible?
2) Theorem 1 holds for a single layer. Can anything be conjectured about the capabilities of more than one layer (other than anecdotal examples)? Shades of 'perceptrons can only compute linearly separable classifiers' here.
3) How do these results compare with those of Lake, B. M., & Baroni, M. (2023). Human-like systematic generalization through a meta-learning neural network. Nature, 623(7985), 115-121., where the network is able to represent e.g. `tiptoe backwards around a cone` given `tiptoe`, `walk backwards`, and `walk around a cone`, and 3 layers are used.
4) MLPs are universal function approximators, and $\Phi$ could include an MLP. Could the MLP approximate $f(g(x))$? I'm guessing there are known complexity/space bounds here, how do these interact with your arguments and proofs? Would be good to have a line or two on this.

Minor comments/typos
- p2: the the father
- p2: idexicals -> indexicals
- p3: "That is, the LLM generates outputs even when the probability of the predicted token is low. When this intensifies, the model will generate a low-likelihood series of tokens, resulting in an unreliable output." Why should the output be unreliable if the series of tokens is low-likelihood? I think the problem here is more that incorrect answers are higher likelihood given the cropus, which is why the output becomes unreliable.
- p4: Why is the token of the query `John`? why isn't it `?`?
- p4: "Notice that the number $N$ of input tokens is a small multiple of the domain size of the functions." Why is this important? Not referred to in the rest of the paper.
- p4: Does the notation $[n]$ mean a set of cardinality $n$? Please add this in.
- p5: "Faye supplies the token strings associated with function f — say, the value of f applied to 0 is 3, etc." What goes into the etc.? All of the values of f from 0 to n-1?
- p9: CoT [needs a] prompt of length $\Omega(\sqrt{N})$ - is this correct? is it the same $N$ as on p4?

**Reasons To Accept:**

- Very interesting topic

- Very important topic

- Likely to form a good basis for further theory in the area

- Mostly well written

**Reasons To Reject:**

- I have a bunch of questions that are outlined in the Questions to Authors. This is not an area of expertise for me so I am unsure how critical the questions I raise are.

- Currently limited to a single transformer layer, would be good to have a conjecture about the extent to which the results generalise to >1 layer.

---

> ### Author Rebuttal · Authors · 2024-05-30
>
> Many thanks for your positive feedback; we address your concerns below.
>
> > Theorem 1 and positional embeddings.
>
> Our proof takes positional encoding into consideration, and holds in its presence. At a high level, our proof shows that in order to compute $\frac{F+G+X}{F’+G’+X’}$, the total number of bits transferred is at most H(d+1)p (i.e., the total number of bits for F, G, F’,G’), while the composition task requires n\log(n) bits.  Positional encoding does not affect this argument.
>
> >  continuous interval [1,n]
>
> A general function over a continuous domain such as [1,n] would take infinitely many bits.  The maximum meaningful domain size for Transformers is the total number of tokens at play plus the prompt length.  In the composition task, the prompt consists of the description of the function, hence the prompt length is an upper bound.
>
> > more than one layer?
>
> Thanks for your suggestion, this is an important question. Indeed, we conjecture that any constant layer Transformer could not resolve the iterated composition task (chain of thought Section has a formal definition). We believe this is true is because it is hard to communicate exactly token embeddings between layers.  However, we have not yet found a rigorous way to capture this intuition.
>
> > Lake, B. M., & Baroni, M. (2023).
>
> Thank you. Empirically, it solves certain instances of composition.  We prove that not all instances of composition problems can be solved by single layer transformers — and conjecture, by multi-layer transformers as well.  Also, observation 1 and its impossibility consequences apply for multiple layers.  Empirical results on sequential composition show problems in SOTA LLMs (Dziri et al, 2023).
>
> > Universal function approximators..
>
> The universal approximation theorem of DNN applies when the number of neurons scales exponentially with respect to the input. If we allow an exponential size Transformer (e.g. the precision or the hidden dimension is exponentially large), then we have similar universal approximation results.
>
> > p3:  incorrect answers are higher likelihood given the corpus
>
> We agree with you, LLMs fulfill remarkably well their contract to capture the probabilistic linguistic structure of the corpus, and this means that making mistakes is to be expected, in fact required.  We make a similar point at the end of our introduction, and Kalai and Vempala 2023 in our refs makes this point more precisely.
>
> Thanks for the writing suggestions, we will incorporate them.

---

> > ### Author Response · Authors · 2024-06-04
> >
> > We thank the reviewer for the valuable comments. Since it is close to the end of author-reviewer discussion, please let us know if we have properly addressed your questions.

---

> > > ### Comment · Reviewer_7NpS · 2024-06-04
> > >
> > > Thanks for your response. I am still a little confused what 'domain' means in your context. Are you saying that you conceptualize the Transformer as a function from prompt to output? So I guess this would make the domain the space $vocab^{prompt_length}$. But then you limit the domain for a particular case by counting the domain as e.g. the number of facts in the prompt, or similar? In any case, it would be really helpful to have a formal definition of 'domain' in the paper, as this concept is critical to the paper.
> > >
> > > Please can you clarify:
> > > - p4: Why is the token of the query `John`? why isn't it `?`?
> > > - p4: "Notice that the number $N$ of input tokens is a small multiple of the domain size of the functions." Why is this important? Not referred to in the rest of the paper.
> > > - p5: "Faye supplies the token strings associated with function f — say, the value of f applied to 0 is 3, etc." What goes into the etc.? All of the values of f from 0 to n-1?
> > > - p9: CoT [needs a] prompt of length $\Omega(\sqrt{N})$ - is this correct? is it the same $N$ as on p4?
> > >
> > > Thank you!

---

> > ### Author Response · Authors · 2024-06-05
> >
> > Thanks for your response; we address your questions below.
> >
> > > Thanks for your response. I am still a little confused what 'domain' means in your context. Are you saying that you conceptualize the Transformer as a function from prompt to output? So I guess this would make the domain the space . But then you limit the domain for a particular case by counting the domain as e.g. the number of facts in the prompt, or similar? In any case, it would be really helpful to have a formal definition of 'domain' in the paper, as this concept is critical to the paper.
> >
> > Thanks for asking us to clarify this.  Our paper shows that, when the prompt involves the definition of one or two functions, plus a particular query, eg,
> > “the value of the function at 0 is 3, … , the value of the function at 9 is 2, what is the value of the function applied to the result of the function applied to 3?”
> > then a Transformer layer is likely to hallucinate.  We denote by N the number of tokens in the prompt of the Transformer (in this case $N$ is about 110), and by $n$ the size of the domain of the function defined by the prompt (in this case n is 10).  Obviously, $N=O(n)$ and $n=O(N)$.
> > We hope this clarifies the issue.
> >
> >
> > > p4: Why is the token of the query John? why isn't it ??
> >
> > Thanks for pointing this typo out. The token of the query is “?” or potentially an EOS token. We will make the correction in the paper.
> >
> >
> > > p4: "Notice that the number of input tokens is a small multiple of the domain size of the functions." Why is this important? Not referred to in the rest of the paper.
> >
> > It is not important.  It means that, for the rest of this argument, we can use $n$ and $N$ interchangeably without harm to the asymptotic behavior.
> >
> > > p5: "Faye supplies the token strings associated with function f — say, the value of f applied to 0 is 3, etc." What goes into the etc.? All of the values of f from 0 to n-1?
> >
> > Yes.
> >
> > > p9: CoT [needs a] prompt of length $\sqrt{N}$  - is this correct? Is it the same N as on p4?
> >
> > CoT generates a sequence of reasoning steps before outputting the answer. Theorem 2 proves that CoT needs to generate at least $\sqrt{N}$ reasoning steps to solve iterated function composition. This is the same $N$ as p4.

---

> > > ### Comment · Reviewer_7NpS · 2024-06-05
> > >
> > > Ok, thanks, I will raise my score slightly

---

### Official Review · Reviewer_Cz42 · 2024-05-13

**Rating:** 7
**Confidence:** 2
**Ethics Flag:** 1

**Summary:**

This paper describes a number of theoretical limitations of transformer models, from the perspective of computational complexity theory. A key result is that transformers cannot represent arbitrary compositions of functions, unless P = NP (this is because the former have O(n^2) runtime complexity whereas some function compositions are NP-hard).

**Questions To Authors:**

It is known that theoretically, DNNs can represent any real-valued function (even a shallow DNN can, provided you arbitrarily increase its number of neurons). It would be interesting to see whether this also holds for transformers, or whether there is a limitation here.

But in general, it would be nice if you think a little on any potential implications to their practical use.

**Reasons To Accept:**

- Interesting mathematical paper.
- Limitations hold provided we abide by the assumptions made by the authors.

**Reasons To Reject:**

- This paper seems a bit too theoretical for the venue, and would be better received at a logic or computability conference, rather than at an NLP venue.
- It is not clear what class of functions the paper targets: real-valued functions? integer-based functions? first-order queries? It is difficult to understand the scope of the results.
- The example given describes a kind of join, but the database is only incompletely specified (the prompts are lacking any intensional knowledge as people from logic programming would put it).
- DNNs do not reason symbolically, but rather **approximate** (with reasonable accuracy/low error) tasks via tensor multiplication, and learning from individual examples w. loss functions and back-propagation. They can't by definition be used to "express" arbitrary functions. I think that the authors need to make clear that the theoretical limitations they discover don't necessarily have a negative impact on their practical applications. Why should I care that they can't express or fully solve function composition in your definition if they can do it with > 90% accuracy?

---

> ### Author Rebuttal · Authors · 2024-05-30
>
> Thanks for the positive feedback; we address your concerns below.
>
> > This paper seems a bit too theoretical for the venue, and would be better received at a logic or computability conference, rather than at an NLP venue.
>
> Thanks for your suggestion. We will try our best to rewrite our paper so as to make our work accessible to the general audience. Our work concerns the theoretical limitations of Transformer architecture, which we believe is well-aligned with the scope of COLM.
>
> > It is not clear what class of functions the paper targets: real-valued functions? integer-based functions? first-order queries? It is difficult to understand the scope of the results.
>
> The composition task considered in our paper deals with discrete domains, such as persons and dates.  We believe we define this quite clearly, and your next question captures our intention well.
>
> >The example given describes a kind of join, but the database is only incompletely specified (the prompts are lacking any intensional knowledge as people from logic programming would put it).
>
> Our scope is Transformers and their prompts.  The database analogy works only up to a point.  A prompt can be, as you say, only an incomplete database.
>
> > DNNs do not reason symbolically, but rather approximate...
>
> Our work indeed takes approximation into consideration. We believe that the failure to answer queries such as this and the resulting hallucinations are a problem.  For example, in Theorem 1, if Hdp << nlog(n) (Hdp is roughly the size of one Transformer layer, n is the prompt length), then one-layer Transformer fails to solve the composition task with probability at least 0.99.
>
> > It is known that theoretically, DNNs can represent any real-valued function (even a shallow DNN can...
>
> The universal approximation theorem of DNN requires that the number of neurons can scale exponentially with respect to the input. If we allow exponential size Transformer (e.g. the precision or the hidden dimension is exponentially large), then we have similar universal approximation results.
>
> > But in general, it would be nice if you think a little on any potential implications to their practical use.
>
> Thanks for your suggestions. Our work was motivated by, and is meant to explain, empirical research (e.g. Dziri et al. 2023 in our citation, among many others among our citations) and we are happy to explore more practical implications of our results.

---

> > ### Author Response · Authors · 2024-06-04
> >
> > We thank the reviewer for the valuable comments. Since it is close to the end of author-reviewer discussion, please let us know if we have properly addressed your questions.

---

> > > ### Comment · Reviewer_Cz42 · 2024-06-05
> > > **Aknowledgement**
> > >
> > > Yes, thank you!

---

### Decision · Program_Chairs · 2024-07-10

**Decision:**

Accept

**Comment:**

Some reviewers argue in favor of acceptance. The authors made an effort to address delicate questions during the response period.
Towards Accept.